



# Strong particle production and condensational growth in the upper troposphere sustained by biogenic VOCs from the canopy of the Amazon Basin

Yunfan Liu[1], Hang Su[2], Siwen Wang[1], Chao Wei[2], Wei Tao[2], Mira L. Pöhlker[2,6,7], Christopher Pöhlker[2], Bruna A. Holanda[2], Ovid O. Krüger[2], Thorsten Hoffmann[5], Manfred Wendisch[7], Paulo Artaxo[8], Ulrich Pöschl[2], Meinrat O. Andreae[3,4], and Yafang Cheng[1]

[1] Minerva Research Group, Max Planck Institute for Chemistry, 55128 Mainz, Germany

[2] Multiphase Chemistry Department, Max Planck Institute for Chemistry, 55128 Mainz, Germany

[3] Max Planck Institute for Chemistry, 55128 Mainz, Germany

[4] Scripps Institution of Oceanography, University of California, San Diego, La Jolla, CA, USA

[5] Institute of Inorganic and Analytical Chemistry, Johannes Gutenberg University Mainz, Mainz, Germany

[6] Experimental Aerosol and Cloud Microphysics Department, Leibniz Institute for Tropospheric Research,

Leipzig, Germany

[7] Faculty of Physics and Earth Sciences, Leipzig Institute for Meteorology, Leipzig University, Leipzig, Germany

[8] Institute of Physics, University of São Paulo, São Paulo, Brazil

* *Correspondence to*: *yafang.cheng@mpic.de*

**Abstract.** Nucleation and condensation associated with biogenic volatile organic compounds (BVOCs) are important aerosol formation pathways, yet their contribution to the upper tropospheric aerosols remains inconclusive, hindering the understanding of aerosol climate effects. Here, we develop new schemes describing
these organic aerosol formation processes in the WRF-Chem model and investigate their impact on the abundance of cloud condensation nuclei (CCN) in the upper troposphere (UT) over the Amazon Basin. We find that the new schemes significantly increase the simulated CCN number concentrations in the UT (e.g., up to ~400 cm$^{-3}$ at 0.52% supersaturation) and greatly improve the agreement with the aircraft observations. Organic condensation enhances the simulated CCN concentration by 90% through promoting particle growth, while
organic nucleation, by replenishing new particles, contributes an additional 14%. Deep convection determines the rate of these organic aerosol formation processes in the UT through controlling the upward transport of biogenic precursors (i.e., BVOCs). This finding emphasizes the importance of the biosphere-atmosphere coupling in regulating upper tropospheric aerosol concentrations over the tropical forest and calls for attention to its potential role in anthropogenic climate change.



# 1 Introduction

Atmospheric aerosol particles can influence the Earth's climate by acting as cloud condensation nuclei (CCN), among other pathways. The CCN residing in the upper troposphere (UT), which have been repeatedly observed in a substantial amount over the globe (e.g., Minikin et al., 2003; Andreae et al., 2018, and references therein; Williamson et al., 2019), not only constitute an important aerosol source for the lower troposphere

(Wang et al., 2016a; Williamson et al., 2019), but also can directly be activated into cloud droplets through in-cloud activation and thus alter the cloud properties (Paluch and Knight, 1984; Fan et al., 2018). However, the formation mechanisms for the upper tropospheric CCN are poorly understood, which impedes their representation in models and the assessment of their climate effects (Heald et al., 2011; Watson-Parris et al., 2019).

Large concentrations of cloud active aerosol particles were detected in the UT over the Amazon by aircraft observations during the ACRIDICON-CHUVA campaign (Wendisch et al., 2016; Andreae et al., 2018). Chemical analysis demonstrated that their composition is dominated by organic compounds with signatures of secondary organic aerosol (SOA) related to the oxidation of biogenic volatile organic compounds (BVOCs; Schulz et al., 2018), yet, detailed processes driving the biogenic SOA formation remain inconclusive. Generally,

two mechanisms may promote the CCN production from biogenic SOA in the UT. The first relates to organic new particle formation (NPF), where aerosol particles can form out of nucleation of highly oxygenated molecules (HOMs) oxidized from biogenic organic vapors such as α-pinene and β-pinene (Burkholder et al., 2007; Kirkby et al., 2016) and subsequently grow to larger sizes. The pure organic NPF can notably affect the atmospheric CCN budget in the planetary boundary layer (PBL; Gordon et al., 2016). Alternatively, if there are

enough preexisting fine particles in the UT from transport or inorganic nucleation, the condensation of low volatile organic compounds (LVOCs) produced by BVOCs oxidation onto the preexisting particles can also increase the CCN number (D'Andrea et al., 2013). However, a quantitative assessment of the BVOC-driven nucleation and condensation processes is lacking (Tröstl et al., 2016; Williamson et al., 2019). To what extent these two processes account for the CCN production in the Amazon UT, and whether these processes and CCN

formation proceed in the UT or if CCN form in the lower troposphere and then are transported upwards, is not known.

    Motivated by these questions, this study implements the laboratory-based organic nucleation (HOMs-induced nucleation) and condensation processes into the WRF-Chem model and conducts simulations to quantify the CCN production from these BVOCs-driven SOA formation pathways in the Amazon UT. We

explore the upper tropospheric biogenic SOA formation mechanisms in terms of the involved atmospheric physical and chemical processes and on a diurnal variation scale.

# 2 Results

## 2.1 Simulation of number concentration of CCN and total aerosol particles in the upper troposphere

    The organic nucleation mechanism in this study focuses on pure organic nucleation, including neutral and

ion-induced processes, triggered by HOMs from α-pinene and β-pinene oxidation (Kirkby et al., 2016), as it was found dominant among organic nucleation pathways (Zhu et al., 2019). The organic condensation mechanism addresses the dynamic condensation of LVOCs oxidized from α-pinene, β-pinene, and isoprene (Mann et al.





2010) as well as HOMs. For an accurate representation of HOMs concentrations, we adopted the Common
Reactive Intermediates gas-phase Mechanism (CRIMech) scheme (Jenkin et al., 2008) with an explicit
description of α-pinene and β-pinene oxidation and calculated the HOMs concentration dynamically, which
circumvented the uncertainties related to species approximation in other chemical schemes (Riccobono et al.,
2014; Zhu et al., 2019) and the bulk assumption of an equilibrium state of HOMs (Gordon et al., 2016; Tröstl et
al., 2016), respectively. The temperature effects on nucleation rate and LVOC yields were included in the model
according to a combination of nucleation theory (Yu et al., 2017) and experimental results (Sahhaf et al., 2008).
Details of the model description can be found in Appendix Sections A1.1 and A1.2.

To disentangle the organic nucleation effect from the organic condensation influence, we performed the
following sensitivity simulations:

- BASE, the default WRF-Chem simulation with $H_2SO_4$-$H_2O$ binary nucleation and without biogenic
  nucleation or condensation;
- CTRL, in which both the newly developed nucleation and condensation modules were added;
- OCD, which only added the organic condensation;
- BNUnoT and OCDnoT, which excluded the temperature effect on the nucleation rate and the LVOC
  yields, respectively;
- NoOH which was based on CTRL but without the HOMs formation from OH oxidation.

Settings for all scenarios are summarized in Table A3. The simulations were conducted for two nested domains
covering the Amazon with a horizontal resolution of 75 km and 15 km, respectively (Fig. A1) from 1 September
to 1 October 2014. Aircraft measurements of aerosol number concentration profiles reaching up to 15 km
altitude, close to the tropopause (18 km, Wendisch et al., 2016), sampled during the ACIRIDICON-CHUVA
campaign (Wendisch et al., 2016; Andreae et al., 2018) were used to evaluate the model results. Details of the
model configuration and observation dataset are documented in Appendix Section A1.3–A2.

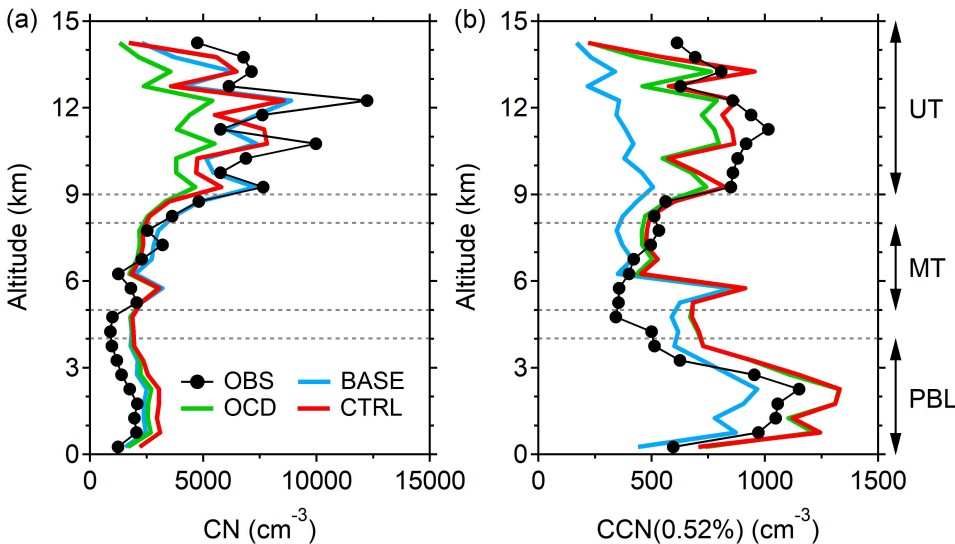

**Figure 1.** Comparison of observed and simulated number concentrations of (a) condensation nuclei (CN, total aerosol population) with diameters above 20 nm, and (b) cloud condensation nuclei (CCN) at 0.52% supersaturation. The aerosol concentrations are at standard temperature and pressure (STP; 273.15 K and 1000 hPa). Planetary boundary layer (PBL), middle troposphere (MT), and upper troposphere (UT) are defined as the altitude range of 0–4 km, 5–8 km, 9–15 km, respectively. The standard deviations of the observations are

provided in Table A4.

Figure 1 shows the average profiles of the number concentration of aerosol particles (also called condensation nuclei, CN) with a diameter above 20 nm and of the CCN at a supersaturation of 0.52% from aircraft observation (Fig. A1) and model simulations. The size of the CCN is approximately 90 nm in diameter,

calculated according to the algorithm of Su et al. (2010; Section A3). Compared with the observations, the BASE case appears to reproduce the vertical distribution of CN in general (Section A3), but the CCN concentration in this simulation shows noticeable biases, especially in the UT (9–15 km) where the model underestimates the observed CCN number by up to 500 cm$^{-3}$ (58%). Considering the different size ranges in which CN and CCN reside, the large model underestimation in CCN may suggest insufficient growth of the

smaller particles in the UT.

When adding the particle growth from the LVOCs condensation into the model (i.e., the OCD case), the simulated CCN number in the UT rises notably, with an increase of 310 cm$^{-3}$ (about 90% relative to the BASE case; Table A4). However, the larger particles from the condensation growth meanwhile deplete the nano-sized particles, causing a dramatic drop in CN number concentrations from BASE to OCD. Though the non-cloud-

resolving resolution of the simulations may cause an excessive mixing of ultrafine particle-laden fresh cloud outflows and their surrounding airmasses (Andreae et al., 2018) and thus aggravate the particle scavenging, the considerable underestimation of the averaged CN number under a reasonable condensation growth in OCD is strongly indicative of some missing NPF mechanisms (Zhu et al., 2019; Zhao et al., 2020). As expected, by further taking into account the organic nucleation (i.e., the CTRL case), the simulated CN number

concentrations are enhanced substantially (2100 cm$^{-3}$, over 50%; Table A4) relative to the OCD case and in markedly better agreement with the observation, while the CCN number concentrations in the model show a relatively weaker increase (90 cm$^{-3}$, about 14%; Table A4). Thus, in total, both the BVOCs-driven organic nucleation and condensation play important roles in maintaining the particle population and size distribution in the UT (Fig. A7). The HOMs nucleation effectively increases the CN number by replenishing new nano-sized

particles, yet its contribution to the CCN, which are mainly in accumulation mode, is relatively limited. In contrast, the organic condensation causes efficient particle growth and, therefore, greatly enhances the CCN population.

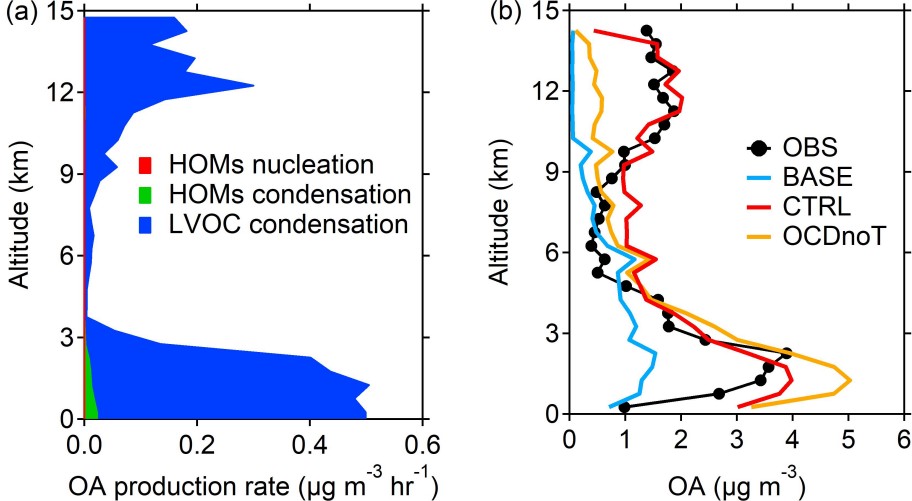

**Figure 2.** Profiles of (a) contribution to organic aerosol (OA) mass from secondary organic aerosol production processes and (b) observed and simulated OA mass averaged along all the flight tracks. The aerosol concentrations are at STP.

       Consistent with the CCN behavior, organic aerosol (OA), the dominant aerosol component over the

simulated region and period (Andreae et al., 2018), is also underestimated in the UT in default WRF-Chem (BASE), but improved close to observation when the biogenic SOA formation is included (Fig. 2b). The condensation of LVOCs plays a predominant role in the OA mass production among all processes, while the other two formation pathways, especially the HOMs nucleation, contribute little (Fig. 2a). This also explains why the organic condensation can cause profound particle growth while the HOMs nucleation works mainly to

increase the number of small particles. The OA production from the SOA processes in the UT shows a similar vertical pattern to that of the OA mass (Fig. 2b), implying local origins of the upper tropospheric CCN.

**2.2 Factors influencing organic-driven particle formation and growth in the upper troposphere**



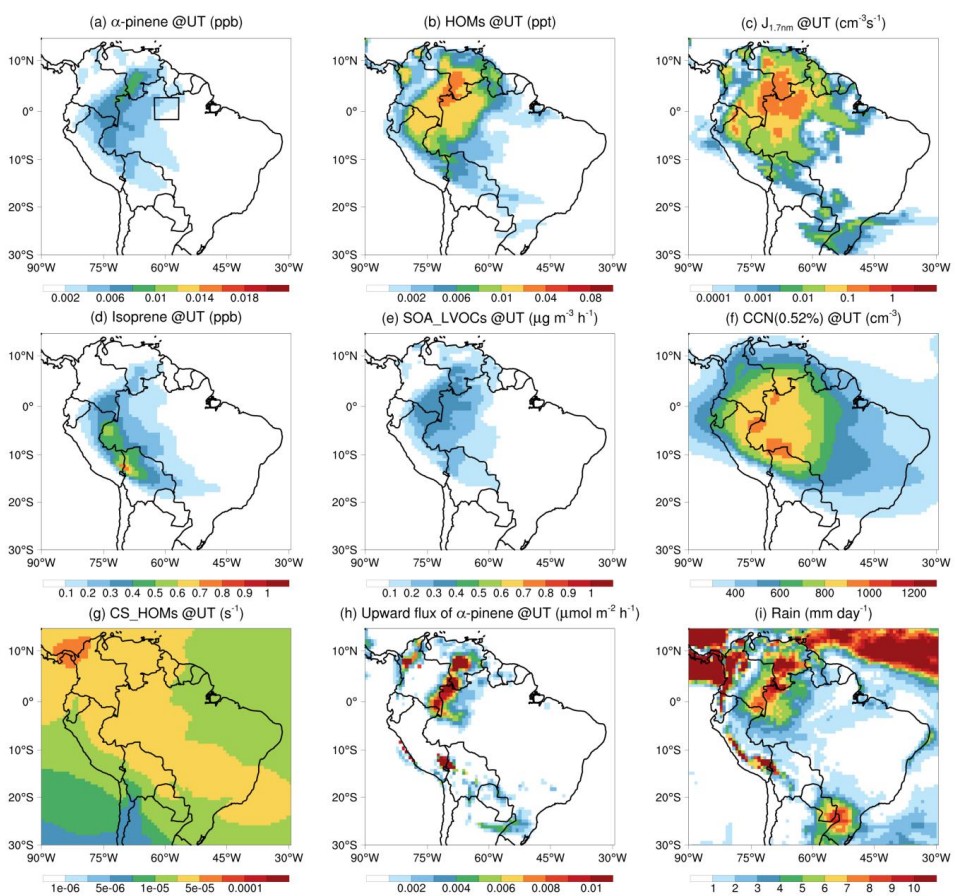

**Figure 3.** Spatial distribution of (a) α-pinene, (b) HOMs, (c) organic nucleation rate, (d) isoprene, (e) SOA production rate by LVOCs, (f) CCN at 0.52% supersaturation, and (g) condensation sink of HOMs in the upper troposphere (UT) averaged over 1 Sep–1 Oct 2014. Also shown are (h) upward α-pinene flux to the UT and (i) precipitation rate averaged over the same period. The black rectangle in (a) denotes the region of the Central Amazon for further analysis in this study.

Figure 3 shows the horizontal distributions of the monthly mean biogenic organic precursors (α-pinene and isoprene; α-pinene is used here as a surrogate for the organic nucleation precursors) and the HOMs nucleation ($J_{org}$) and LVOCs condensation rates (SOA_LVOCs) in the UT. The organic nucleation and condensation distributions closely follow that of the biogenic precursors (Fig. 3a, c, d, e) but not the oxidants ($O_3$ and OH; Fig. A8), suggesting the upper tropospheric BVOCs concentration as the limiting factor for the organic nucleation/condensation in the Amazon UT. The region with high BVOC concentrations in the UT is different from the α-pinene distribution in the PBL (Fig. A9), but identical to the precipitation pattern as well as the large upward α-pinene flux (Fig. 3i, h), showing a necessary role of deep convection transport in the BVOCs availability in the UT.



165       The $J_{org}$ can reach over 0.1 cm$^{-3}$ s$^{-1}$ in the UT (Fig. 3) while the PBL (0–4 km) and middle troposphere (MT; 5–8 km) show low values (Fig. A9–A10), even though the α-pinene concentration in PBL is a magnitude larger than in the UT. Such high upper tropospheric $J_{org}$ is favored by not only the low sink of HOMs (CS) but also the low temperature in the UT (Fig. 3g). When the temperature dependence of $J_{org}$ (Section A1.2) is not considered, the $J_{org}$ in the UT is much lower than in the model run with the temperature effect (Fig. A11). The overall

magnitude of $J_{org}$ is lower than simulated previously in the Amazon (Zhu et al., 2019), possibly due to the consideration of the ion sink in this study. For the SOA production from LVOCs condensation, the bulk assumption of the LVOC yields used previously in the boreal forest (Scott et al. 2014) fails to reproduce the observed OA mass due to different conditions in the tropics (Section A4). A temperature-dependent correction of LVOC yields based on laboratory experiments (Saathoff et al., 2009) is necessary for correcting the OA

simulation bias associated with the bulk LVOC yields assumption (Fig. 2b). The low temperature in the UT also serves as a favorable condition for the SOA production from LVOCs condensation.



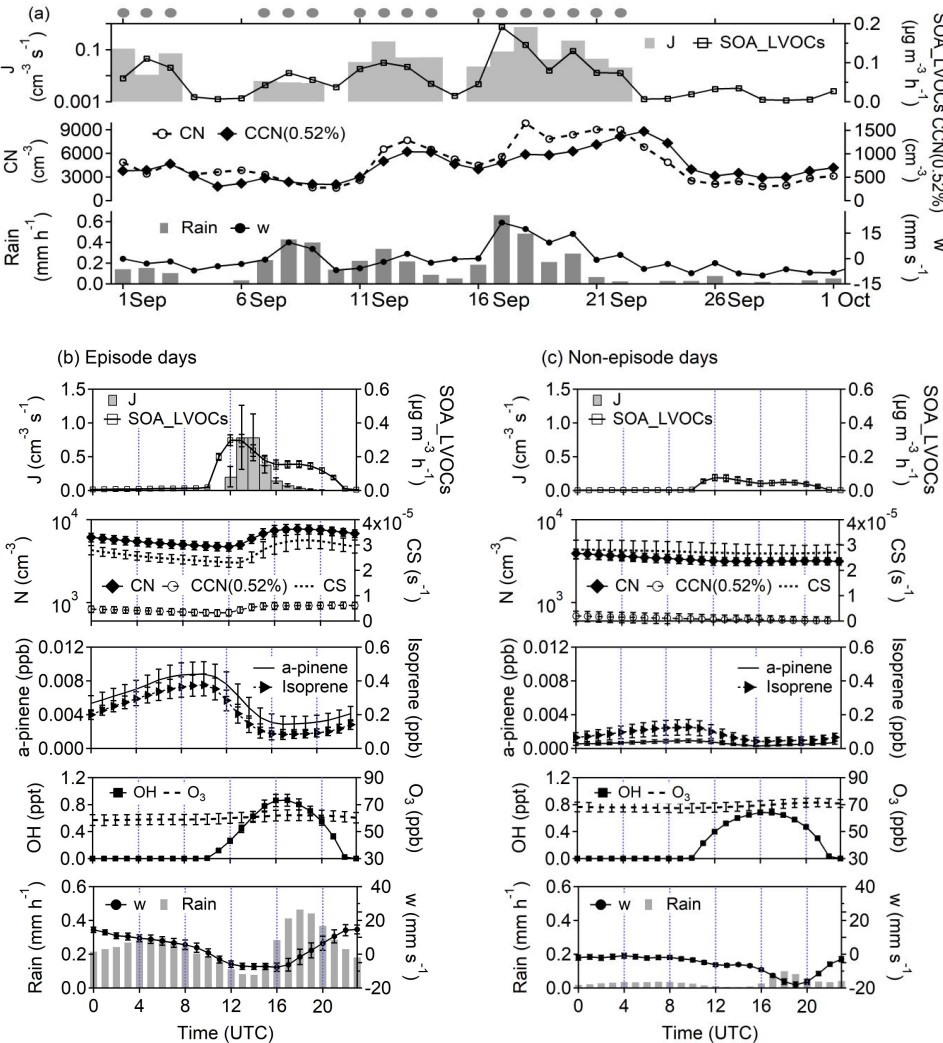

**Figure 4.** Modeled temporal variation of the rate and precursors of organic nucleation and condensation as well as meteorological variables. (a) Daily variation of regionally averaged nucleation and condensation rates, number concentration of CN and CCN at 0.52% supersaturation (CCN(0.52%)), vertical wind at 8 km (w), and precipitation. The dots above the figure mark the upper troposphere (UT) biogenic SOA episodes. The diurnal patterns of nucleation and condensation rate, number concentration of CN and CCN(0.52%), condensation sink of HOMs, and mixing ratio of α-pinene, isoprene, $O_3$, and OH average in the UT as well as the vertical wind at 8 km and precipitation for the (b) UT biogenic SOA episode days and (c) non-episode days. The analyzed region is indicated by the black rectangle in Fig. 3a. The bars denote the standard error.

As the biogenic SOA formation in the UT exhibits prominent daily variation, we defined days with $J_{org}$ greater than 1e-3 $cm^{-3}$ $s^{-1}$ and simultaneous condensational SOA production rate larger than 0.05 μg $m^{-3}$ $h^{-1}$ as UT biogenic SOA episodes (Fig. 4a) to identify their characteristics. The biogenic SOA episodes in the UT emerge





with stronger convection where strong vertical wind effectively transports biogenic precursors to the UT and fosters SOA formation (Fig. 4b). In contrast, the biogenic SOA formation in the PBL tends to be suppressed by deep convection (Fig. A14) due to the restrained emission and oxidation of BVOCs by reduced incident solar
radiation in cloudy weather (Fig. A13) as well as the BVOCs dilution near the surface by low-BVOCs air from the MT.

The organic nucleation and condensation rate demonstrate a clear diurnal cycle, with significant occurrence in the daytime. As the oxidant in the HOMs and LVOCs production, OH has reaction rates several orders faster than $O_3$ (Section A1.2; Atkinson et al., 2006), making it the dominant oxidant, which is also evidenced as the
sensitivity study switching off the OH oxidation (NoOH) shows significantly reduced HOMs concentrations (Fig. A15). Thus, the photolysis origin of OH (Fig. 4b–4c) explains the daytime burst of the $J_{org}$ and condensation rate. A detailed mechanism tracking the diurnal variation of the upper tropospheric CCN production can be drawn. The nighttime convective activity as an extension of the former daytime convection contributes to the upper tropospheric accumulation of biogenic precursors. With the onset of solar radiation, the
photochemical reactions start to produce OH, which efficiently oxidizes BVOCs to form HOMs and LVOCs and then triggers the organic nucleation and condensation. The CN and CCN concentrations increase accordingly and reach high levels in the afternoon which is also the typical time for the vigorous development of local convective clouds and thereby favors potential interactions between upper tropospheric CCN and clouds.

## 3 Conclusions

In this work, we developed a new organic nucleation and condensation scheme for the WRF-Chem model to investigate the CN and CCN production in the UT (upper troposphere) by BVOCs-driven SOA formation over a forest canopy region, the Amazon Basin.

The model evaluation against aircraft measurements shows that including the BVOCs-driven SOA formation significantly improves the model agreement with the measured upper tropospheric CCN (at 0.52% supersaturation) number concentrations by elevating the simulated values up to ~400 cm$^{-3}$. Individually, the
organic condensation drives efficient particle growth and promotes the CCN concentration in the UT by about 90%. With the nano-sized particles from $H_2SO_4$-$H_2O$ binary nucleation scavenged under sufficient particle growth, the organic nucleation serves to replenish nano-particles and enhances the upper tropospheric CN and CCN number concentration by over 50% and 14%, respectively. The rates of SOA processes in the UT depend
on deep convection for its vertical transport, and are favored by low condensation sink and temperature at high altitudes.

The considerable CCN production in the UT by BVOCs-driven organic processes underlines the important regulation of biospheric BVOCs on the high-altitude aerosol concentrations. Considering the climate significance of these upper tropospheric aerosols, the biosphere-atmosphere coupling should be emphasized in
the context of climate change, not only for its possible impact on the preindustrial reference state (Gorden et al., 2016), but also for its feedback to climate under the future anthropogenic influence (e.g., deforestation) and climate change.





**Author contributions**

YC designed and led the study. YL and YC conducted the model development. YL performed the model
simulation and analyzed the data. YL, YC, and HS interpreted the results. MOA and UP contributed to
discussing the results. SW, CW, and WT supported model simulation and data visualization. MOA, UP, PA, and
MW coordinated the ACRIDICON-CHUVA observation field campaign. MLP, CP, OOK, and BAH provided
the observation data from HALO for model comparison. YL wrote the paper with input from all coauthors.

**Competing interests**

The authors declare that they have no conflict of interest.

**Acknowledgements**

This study is supported by the Max Planck Society (MPG). YC and YL acknowledge the Minerva Program
of MPG. We also gratefully acknowledge the observational data and data providers from the ACRIDICON-
CHUVA campaign, which was supported by the Max Planck Society (MPG), the German Science Foundation
(DFG Priority Program SPP 1294), the German Aerospace Center (DLR), the FAPESP (Sao Paulo Research
Foundation) grants 2009/15235-8, 2013/05014-0, and 2017/17047-0, and a wide range of other institutional
partners.

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



**Appendix:**

**A1 Model development**

**A1.1 Model description**

The Weather Research and Forecasting model coupled with chemistry (WRF-Chem), version 3.9.1, was utilized in this study to investigate the biogenic organic nucleation and condensation over the Amazon. WRF-Chem is a meteorology-chemistry online coupled regional model, which integrates meteorological, gas-phase, and aerosol fields by calculating transport of chemical species under the same dynamical system as meteorological elements at each time step and considering complicated feedbacks between meteorological field and trace gases and aerosols (Grell et al., 2005).

**Table A1.** WRF-Chem configuration.

| Atmospheric Process | WRF-Chem Option |
|---|---|
| Longwave radiation | RRTMG |
| Shortwave radiation | RRTMG |
| Surface layer | Monin-Obukhov |
| Land surface | RUC |
| Boundary layer | YSU |
| Microphysics | Lin et al. |
| Cumulus | Grell-Devenyi ensemble |
| Gas-phase chemistry | CRIMech |
| Aerosol module | MOSAIC |
| Aqueous-phase chemistry | Fahey and Pandis |
| Photolysis | Fast-J |
| Anthropogenic emissions | EDGAR-HTAP V2 |
| Biogenic emissions | MEGAN |
| Biomass burning emissions | FINNv1.5 |

**Table A2.** Description of aerosol size bins in MOSAIC.

| Default bins | | | | Modified bins | | | |
|---|---|---|---|---|---|---|---|
| Bin name | Low[a] | High[b] | Center[c] | Bin name | Low | High | Center |
| | | | | 01 | 0.6 | 2.4 | 1.2 |
| | | | | 02 | 2.4 | 10 | 5 |
| | | | | 03 | 10 | 39 | 20 |
| 01 | 39 | 156 | 78 | 04 | 39 | 156 | 78 |
| 02 | 156 | 625 | 312 | 05 | 156 | 625 | 312 |
| 03 | 625 | 2500 | 1250 | 06 | 625 | 2500 | 1250 |
| 04 | 2500 | 10000 | 5000 | 07 | 2500 | 10000 | 5000 |

a, b, c: low boundary, high boundary, and geometric mean diameter of the bin in nm, respectively.

The WRF-Chem configurations used are listed in Table A1. We chose the Common Reactive Intermediates gas-phase Mechanism (CRIMech) scheme (Jenkin et al., 2008; Watson et al., 2008) with the KPP interface to simulate the gas-phase chemistry. CRIMech contains up to 112 non-methane volatile organic compounds (VOCs), including α-pinene, β-pinene, and isoprene (Archer-Nicholls et al., 2014; Watson et al., 2008), and adopts 652 chemical reactions involving inorganic species, organic vapors, and their oxidation intermediates, based on the Master Chemical Mechanism (MCM). These chemical settings enable it to directly provide the gas-



phase precursors, i.e. α-pinene, β-pinene and isoprene, and chemical reactions for the biogenic SOA formation.
The Model for Simulating Aerosol Interactions and Chemistry (MOSAIC; Zaveri et al., 2008) was utilized to simulate aerosols, which uses discrete size bins to represent the aerosol size distribution. This study employed the 4-bin version with the size bins distributed as listed in Table A2. Aerosol species in MOSAIC include 5 inorganic ions, i.e. sulfate, nitrate, ammonium, sodium, chloride, and 3 unreactive primary aerosol species, i.e., black carbon (BC), particulate organic matter (POM), and other inorganics (OIN; Fast et al., 2006; Zaveri et al.,
2008). In the WRF-Chem version 3.9.1 MOSAIC aerosol module, the binary nucleation of $H_2SO_4$-$H_2O$ is included (Wexler et al., 1994), while the organic nucleation and condensation are not accounted for. The coagulation process of particles and gas-particle partitioning were parameterized as described in Zaveri et al. (2008). The dry deposition of aerosol is parameterized by the updated resistance-in-series approach of Wesely (1989). The in- and below-cloud wet deposition takes place by activating aerosol from an interstitial state into
cloud-borne particles and calculating the washout rate due to precipitation, respectively (Chapman et al., 2009; Easter et al., 2004). The WRF-Chem model configurations used in this study allow aerosol-cloud interactions, following the way described by Fast et al. (2006) and Chapman et al. (2009); while the aerosol-radiation interactions were not included in the model simulations.

**A1.2 Model improvement**

Based on the aforementioned CRIMech gas-phase chemistry scheme and MOSAIC aerosol scheme, a new module resolving the purely organic nucleation and condensation associated with BVOCs has been added to WRF-Chem, which provides a modeling tool to investigate biogenic SOA formation and its contribution to the upper tropospheric CCN.

  The implementation of the new organic nucleation includes integrating the production of HOMs in the
CRIMech gas-phase chemistry scheme, resetting the sectional bins in the aerosol model, and adding parameterizations of pure organic nucleation mechanisms by HOMs. Specifically, 4 reactions regarding HOM production from the oxidation of α-pinene and β-pinene by $O_3$ and OH were added to the CRIMech mechanism based on reaction coefficients and yields suggested by laboratory experiments (Atkinson et al., 2006). Then, the condensation sink of HOMs was represented according to the algorithm of Kerminen et al. (2004). Instead of
parameterizing the HOMs concentration as the ratio of its production and condensation sink with the assumption that the HOMs were in a thermal equilibrium state (Kirkby et al., 2016; Gordon et al., 2016), the kinetic calculation of the HOMs production and condensation sink in this study enables a more accurate representation of the HOMs concentration.

$$\text{α-pinene} + O_3 = \text{HOMs}: 0.029{\times}1.01 \times 10^{-15} e^{\frac{-732}{T}} \quad\quad\quad \text{(R. A1)}$$

$$\text{α-pinene} + \text{OH} = \text{HOMs}: 0.012{\times}1.2 \times 10^{-11} e^{\frac{444}{T}} \quad\quad\quad \text{(R. A2)}$$

$$\text{β-pinene} + O_3 = \text{HOMs}: 0.0012{\times}1.5{\times}10^{-17} \quad\quad\quad \text{(R. A3)}$$

$$\text{β-pinene} + \text{OH} = \text{HOMs}: 0.0058{\times}2.38 \times 10^{-11} e^{\frac{357}{T}} \quad\quad\quad \text{(R. A4)}$$

  The 4-bin MOSAIC scheme in WRF-Chem addresses aerosols with diameters from 39 nm to 10 μm, which
does not cover the size range of newly formed particles, whose diameters are in the nanometer range (Kulmala, 2003). To explicitly represent the nucleation of vapor into particles, we extended the lower end of the aerosol





size range in the MOSAIC scheme from 39 nm to 0.6 nm by introducing 3 additional size bins whose boundaries are set following the same lognormal size distribution law as the original 4 bins (Table A2). Thus, the newly developed 7-bin MOSAIC scheme can resolve the formation and initial growth of new particles and

assures a high computation efficiency. The 4-bin MOSAIC scheme includes the $H_2SO_4$-$H_2O$ binary nucleation using a thermodynamic equilibrium parameterization where a critical concentration of $H_2SO_4$ is calculated based on air temperature and relative humidity and then the extra $H_2SO_4$ beyond this threshold is parameterized into aerosols centered at 78 nm (i.e., the 39–156 nm bin). The equilibrium method for describing $H_2SO_4$-$H_2O$ binary nucleation is validated for the aerosol size above 10 nm. With extended aerosol size bins in the 7-bin MOSAIC

scheme, we now applied the $H_2SO_4$-$H_2O$ binary nucleation parameterization to the third bin (i.e. 10–39 nm) to not only assure the practical application of this nucleation parameterization (Wexler et al., 1994) but also keep the aerosol size range in agreement with the observations (i.e., starting from 20 nm).

Then, in addition to the existing $H_2SO_4$-$H_2O$ binary nucleation, pure biogenic nucleation mechanisms induced by HOMs were integrated into the MOSAIC module. The mechanisms of organic nucleation were

investigated in CLOUD (Cosmics Leaving OUtdoors Droplets) Chamber experiments (Kirkby et al., 2016), which suggested that the HOMs-induced pure organic nucleation rate ($J_{org}$, unit: cm$^{-3}$ s$^{-1}$) can be represented by the combination of the neutral ($J_n$) and the ion-induced ($J_{iin}$) nucleation rate. The detailed parameterization of the $J_n$ and $J_{iin}$ is as follows:

$$J_{org} = J_n + J_{iin} \qquad \text{(E. A1)}$$


$$J_n = a_1 [HOM]^{a_2 + \frac{a_5}{[HOM]}} \qquad \text{(E. A2)}$$

$$J_{iin} = 2[n_\pm] a_3 [HOM]^{a_4 + \frac{a_5}{[HOM]}} \qquad \text{(E. A3)}$$

where HOMs concentrations are in units of $10^7$ molecules per cubic centimeter and obtained by chemical kinetic calculations as described above; the $a_i$ represent free parameters whose values were suggested by Kirkby et al.

(2016) where $a_1$, $a_2$, $a_3$, $a_4$ and $a_5$ equaled to 0.04001, 1.848, 0.001366, 1.566 and 0.1863, respectively.

$n_\pm$ is the ion concentration produced from radon and galactic cosmic rays and is parameterized as:

$$[n_\pm] = \frac{(k_i^2 + 4\alpha q)^{0.5} - k_i}{2\alpha} \qquad \text{(E. A4)}$$

where q (in cm$^{-3}$ s$^{-1}$) represents the ion-pair production rate and adopts the value of 10 cm$^{-3}$ s$^{-1}$ (Horrak et al.,

2008). $\alpha$ is the ion-ion recombination coefficient (in cm$^3$ s$^{-1}$) and was set to 1.6e$^{-6}$ cm$^3$ s$^{-1}$ here. The ion loss rate, $k_i$, is due to the ion condensation sink (CS) onto aerosols and the ion-induced nucleation:

$$k_i = CS + \frac{J_{iin}}{2[n_\pm]} \qquad \text{(E. A5)}$$

In this study, the condensation sink term was calculated according to the empirical parameterization proposed by Tammet (1991). The HOMs nucleation rate, $J_{org}$, is then modulated by temperature. Unlike the

approximated temperature correction suggested in Dunne et al. (2016), a temperature dependence associated with the Gibbs free energy for forming the critical cluster based on the classical homogeneous nucleation theory (Yu et al., 2017) is used here. We applied temperature corrections to $J_n$ and $J_{iin}$ by multiplying them by a correction factor, $\exp(\Delta G_n/k*(1/T-1/278))$ and $\exp(\Delta G_{iin}/k*(1/T-1/278))$, respectively, and the $\Delta G_n$ and $\Delta G_{iin}$ are



based on smog chamber results (Kirkby et al., 2016). After organic nucleation, the newly formed particles were
added into the smallest bin and underwent subsequent processes such as coagulation, transport, and deposition.

Additionally, a new module addressing the condensation of LVOCs was integrated into WRF-Chem. The LVOCs were oxidation products of α-pinene, β-pinene, and isoprene by $O_3$, OH, and $NO_3$. A yield of 13% for monoterpene oxidation products and 3% for isoprene oxidation products were used in Scott et al. (2014). Laboratory chamber experiments found a temperature dependence of the SOA yield from α-pinene oxidation
(Saathoff et al., 2009). Therefore, instead of constant LVOC yields, temperature-corrected yields based on these laboratory experiment results (Saathoff et al., 2009) were applied in the model here.

**A1.3 Numerical experiment design**

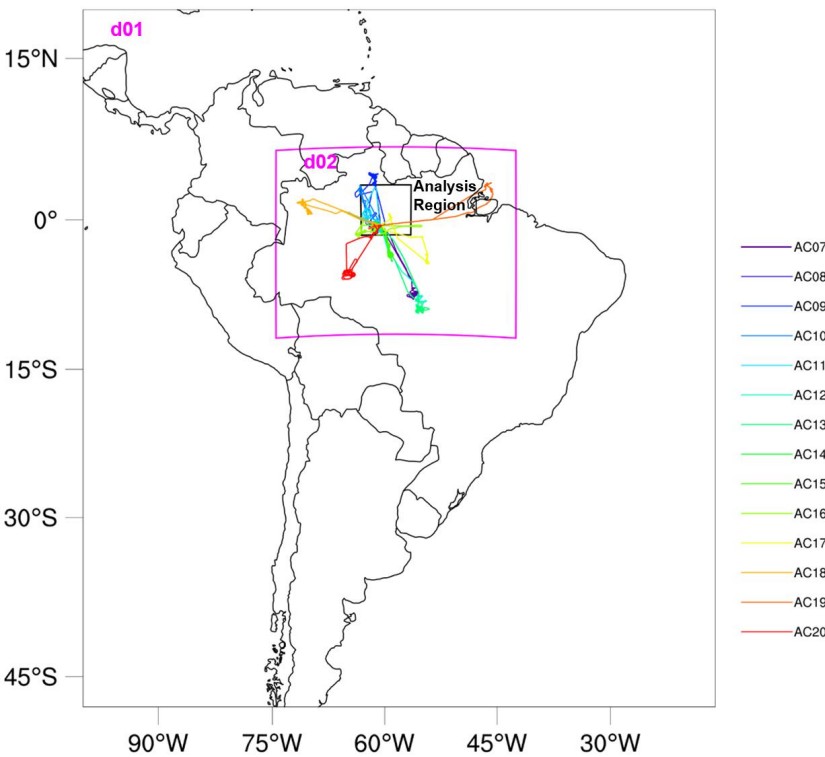

**Figure A1.** Model domain and tracks of flight AC07 to AC20 from the ACRIDICON-CHUVA campaign. The
outer map represents the parent domain with 75 km horizontal grid spacing, and the embedded square shows the extent of the 15 km resolution (d02) domain. The black rectangle is the same as the one marked in Fig. 3a, denoting the region of the Central Amazon for further analysis in this study.

In this study, two nested domains with a horizontal spacing of 75 km and 15 km were set up over South
America (Fig. A1), with Domain1 covering most of the South American continent, while Domain2 is over the Amazon Basin area. Vertical layers of 29 levels extending from the ground surface to a height of 50 hPa were applied for all domains. The initial and boundary meteorological and chemical conditions were from the 6-hour



National Centers for Environmental Prediction (NCEP) Final Analysis (FNL) data and Model for Ozone and Related Chemical Tracers, version 4 (MOZART-4) global chemical transport model output (Emmons et al.,
2010), respectively. The anthropogenic emissions with a resolution of 0.1° × 0.1° were obtained from the global air pollution emission dataset, EDGAR-HTAPv2 (http://edgar.jrc.ec.europa.eu/htap_v2; Janssens-Maenhout et al., 2015). The Fire Inventory from NCAR version 1.5 (FINNv1.5; Wiedinmyer et al., 2011) provided the biomass burning emission, which is updated daily at 1-km resolution. The rise of fire plumes after emission was represented by a plume-rise parameterization (Freitas et al., 2007). The primary organic matter (POM) emission
rate was calculated based on the OC emission by applying a ratio of 1.6 between the mass of POM and OC (Andreae, 2019). The biogenic emissions of NO and volatile organic compounds (VOCs) were generated online by the Model of Emissions of Gases and Aerosols from Nature (MEGAN; Guenther et al., 2012). Among the biogenic VOCs are the precursors, i.e. α-pinene, β-pinene, and isoprene, for the organic nucleation and condensation which are investigated in this study. MEGAN calculated the emission of biogenic gases based on
the United States Geological Survey (USGS) land use category, temperature, and radiation, which were subsequently put into the corresponding chemical species in the CRIMech gas-phase scheme as a source term. The simulation was conducted from 24 August to 1 October 2014, and the first 8 days of the simulation were used as spin up. The comparisons between model outputs and aircraft measurements in Section 2.1 are made with the results from Domain2. A rectangular area focusing on the Central Amazon, as shown in Fig. A1, was
used in the analysis in Section 2.2.

**Table A3.** Experiment design description.

| Experiment identification | Aerosol size range | Inorganic nucleation | Biogenic HOMs-induced nucleation (BNU) | Temperature effect on BNU | LVOCs organic condensation | Temperature effect on LVOC yields |
|---|---|---|---|---|---|---|
| BASE | 0.6 nm–10 μm | Wexler et al. (1994) | No | No | No | No |
| BASEnoNUC | 0.6 nm–10 μm | No | No | No | No | No |
| CTRL | 0.6 nm–10 μm | Wexler et al. (1994) | Yes | Yes | Yes | Yes |
| OCD | 0.6 nm–10 μm | Wexler et al. (1994) | No | No | Yes | Yes |
| BNUnoT | 0.6 nm–10 μm | Wexler et al. (1994) | Yes | No | Yes | Yes |
| OCDnoT | 0.6 nm–10 μm | Wexler et al. (1994) | Yes | Yes | Yes | No |
| NoOH | 0.6 nm–10 μm | Wexler et al. (1994) | Only with HOMs from $O_3$ oxidation but without that form OH oxidation | Yes | Yes | Yes |



To characterize the pure organic nucleation and condensation and investigate its controlling factors, a series
of sensitivity simulations were performed as listed in Table A3. A baseline simulation (BASE) was conducted
based on the default WRF-Chem, except the binary nucleation-generated aerosols were put into the third bin as
described above. Simulation using the improved version of WRF-Chem, CTRL, was conducted, where new
particles can be formed by organic nucleation in addition to the default $H_2SO_4$-$H_2O$ binary nucleation scheme in
BASE, and where the organic condensation process was also taken into consideration. In order to examine the
effect of atmospheric vertical temperature variation on the organic nucleation and condensation growth,
sensitivity simulations were performed using the modified WRF-Chem model, but without temperature
influence on the nucleation rate and the yields of LVOCs, namely BNUnoT and OCDnoT, respectively. For the
purpose of distinguishing the influence from the organic nucleation and the condensation of organics, an
additional sensitivity simulation was made where only the condensation of organics was included in the BASE
case, which was termed OCD. To examine the relative importance of $O_3$ and OH in the HOMs-generating
oxidation reactions, NoOH was conducted based on CTRL but with the HOMs formation from OH oxidation
turned off.

**A2. Data**

The ACRIDICON-CHUVA (ACRIDICON stands for "Aerosol, Cloud, Precipitation, and Radiation
Interactions and Dynamics of Convective Cloud Systems" and CHUVA is the acronym for "Cloud Processes of
the Main Precipitation Systems in Brazil: A Contribution to Cloud Resolving Modeling and to the GPM (global
precipitation measurement)"; Wendisch et al., 2016) campaign was conducted in the Amazon region in 2014. It
aimed at in-depth investigations of the properties of the aerosols and clouds in this area and the explorations of
interactions between aerosols, radiation, clouds, and precipitation. Fourteen flights were operated between 6
September 2014 and 1 October 2014, encompassing comprehensive measurements of meteorology, trace gases,
and aerosols with ceiling heights up to 15 km, close to the top of the troposphere. The measurements of
meteorological parameters (air temperature, relative humidity, and wind speed), $O_3$, total aerosol number
concentration, CCN number concentration, and black carbon and organic aerosol mass conducted on the 14
flights were used in this study. The total aerosol particles, also called condensation nuclei (CN) focus on
aerosols with a diameter above 20 nm. The observed CCN are the CCN at a supersaturation of 0.52%
(CCN(0.52%); Andreae et al., 2018). The flight tracks are shown in Fig. A1. Overviews of the ACRIDICON-
CHUVA campaign and observation are documented by Wendisch et al. (2016) and Andreae et al. (2018).

**A3. model evaluation**



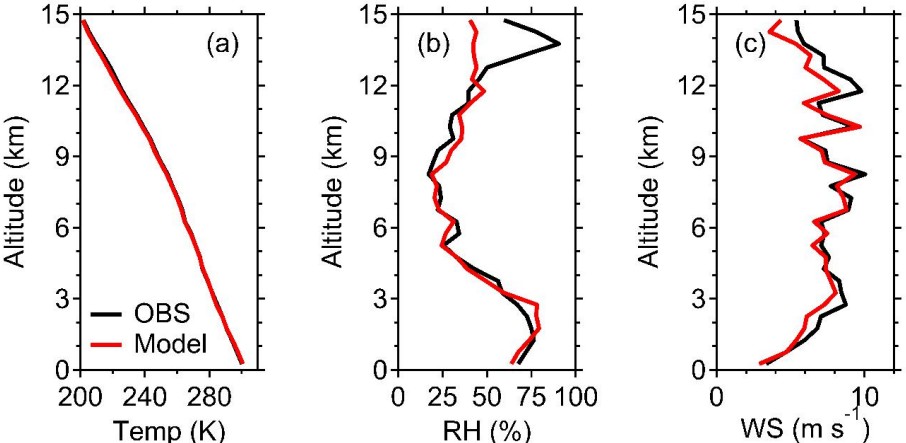

**Figure A2.** Comparison of (a) air temperature (Temp), (b) relative humidity (RH), and (c) horizontal wind speed (WS) averaged from all flight measurements (OBS) and WRF-Chem simulations (Model).

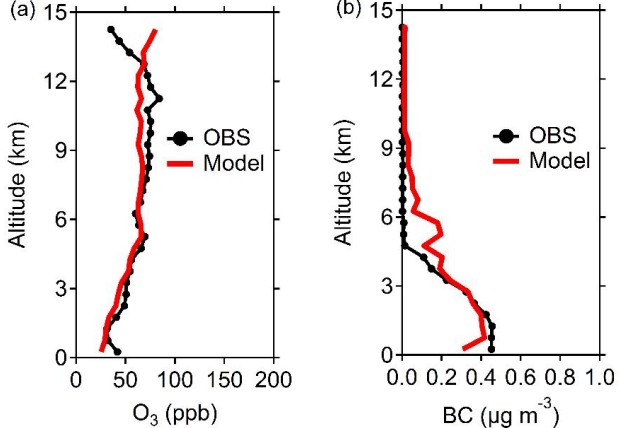

**Figure A3.** Comparison of (a) $O_3$ mixing ratio, and (b) black carbon (BC) mass concentration averaged from all flight measurements (OBS) and WRF-Chem simulations (Model).



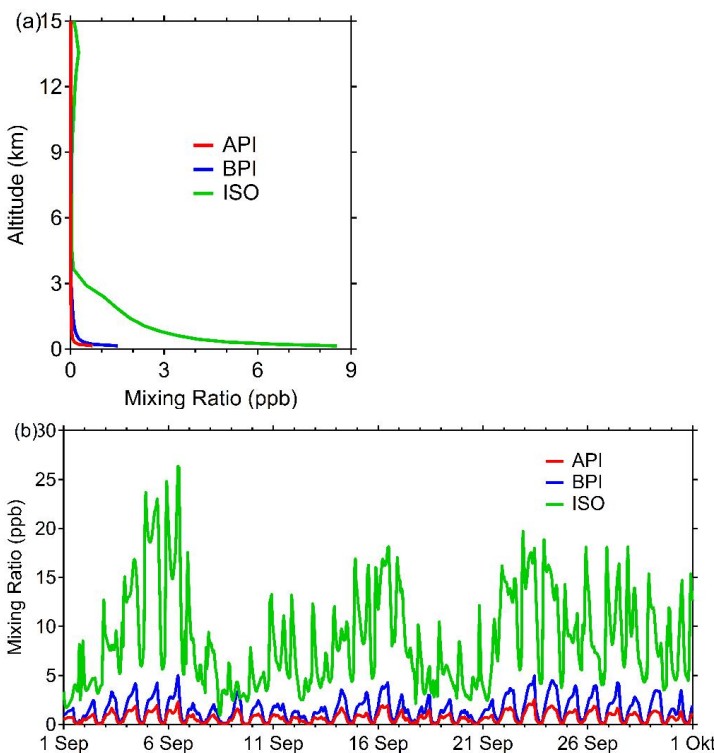

**Figure A4.** Simulated (a) vertical profiles and (b) time series of α-pinene, β-pinene, and isoprene mixing ratios at the location of ATTO.


To compare against the flight observation, the hourly model outputs at the corresponding location of the observed data within the hour were used. The model reasonably reproduced the meteorological conditions (Fig. A2), the $O_3$ vertical distributions, and the black carbon concentrations (Fig. A3), showing its ability to capture the meteorological processes, basic atmospheric chemical processes, and primary aerosol emission and transport

processes. The simulated concentrations of the biogenic organic vapors, α-pinene, β-pinene, and isoprene (Fig. A4) are of comparable magnitude to previous observations (Kuhn et al., 2010), demonstrating a reasonable model simulation of the biogenic emissions.



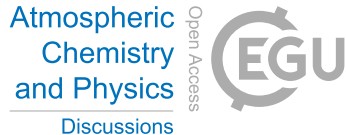

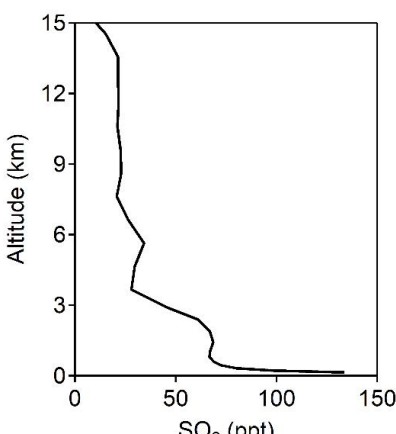

**Figure A5.** Simulated vertical profile of the SO2 mixing ratio at the location of ATTO.


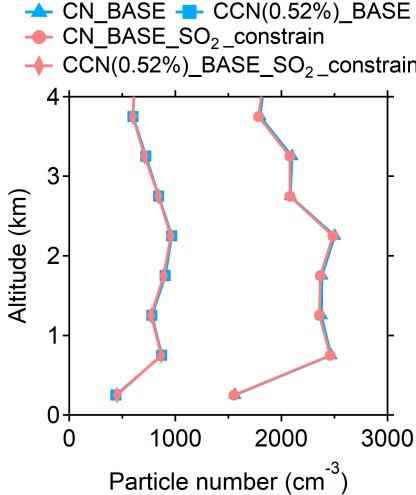

**Figure A6.** Vertical profiles of the simulated number concentrations of CN and CCN at 0.52% supersaturation averaged along the observation trajectories within the planetary boundary layer (PBL).






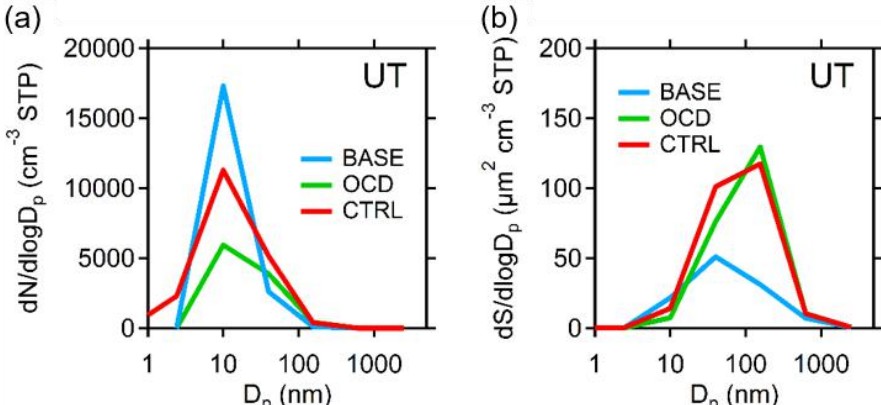

**Figure A7.** Simulated size distributions of (a) particle number concentration, and (b) surface area in the upper troposphere (UT) along the flight tracks.


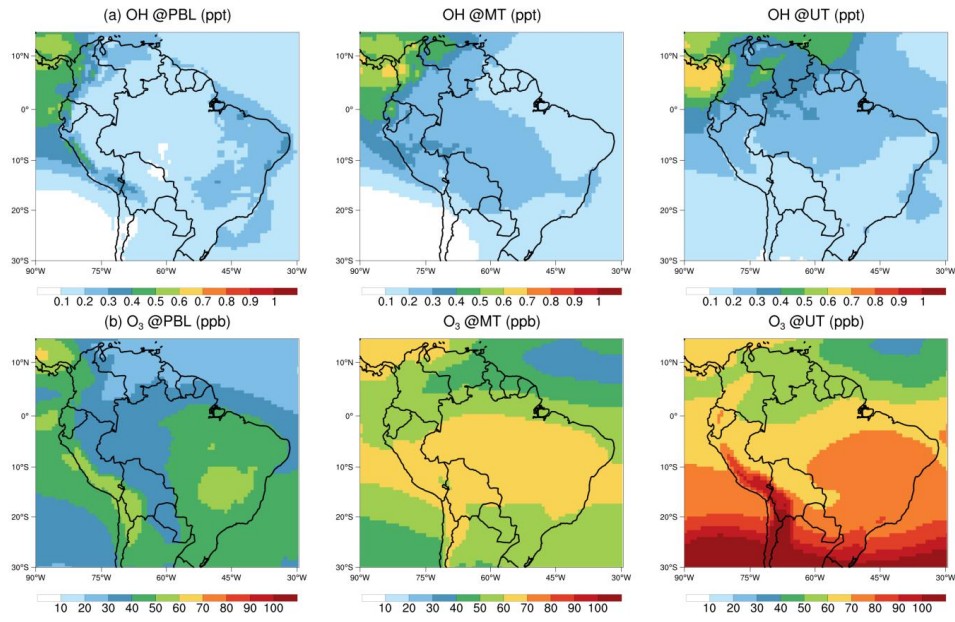

**Figure A8.** Horizontal distribution of (a) OH and (b) O₃ averaged over 1 Sep–1 Oct 2014 from the CTRL case.


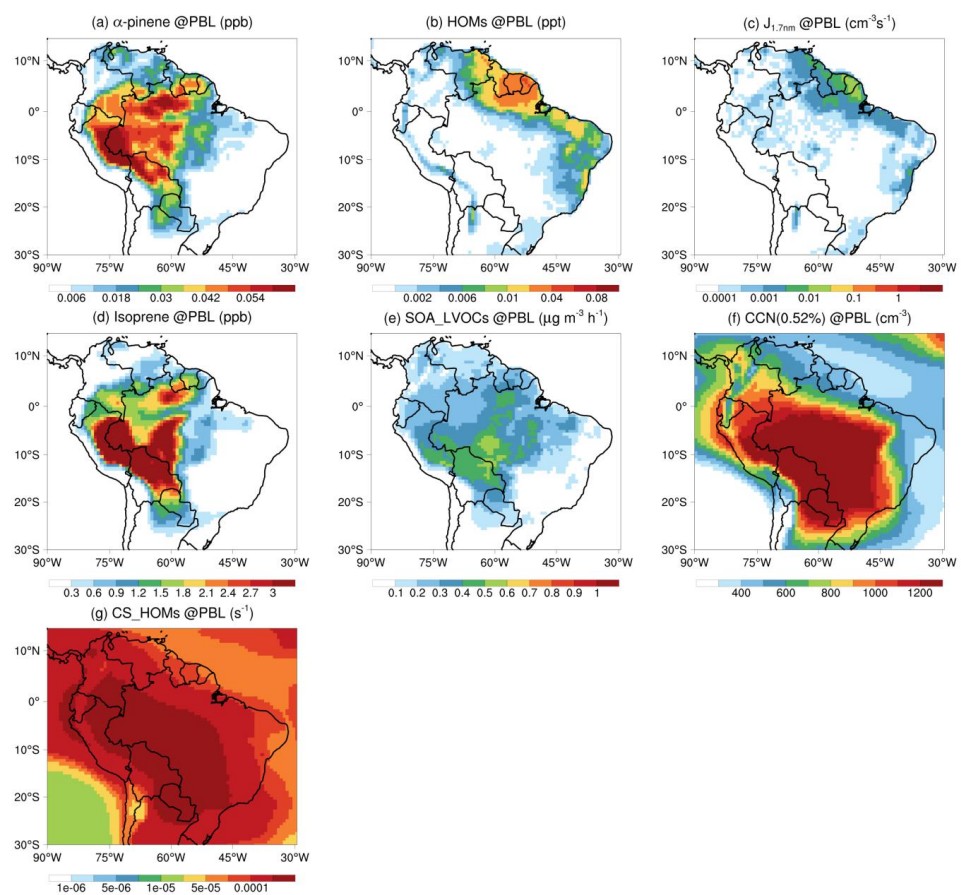

**Figure A9.** Same as Figure 3 (a-g), but for the planetary boundary layer (PBL). Note that the scales in (a) and (d) are 3 times of those in Fig 3a and Fig. 3d, respectively.





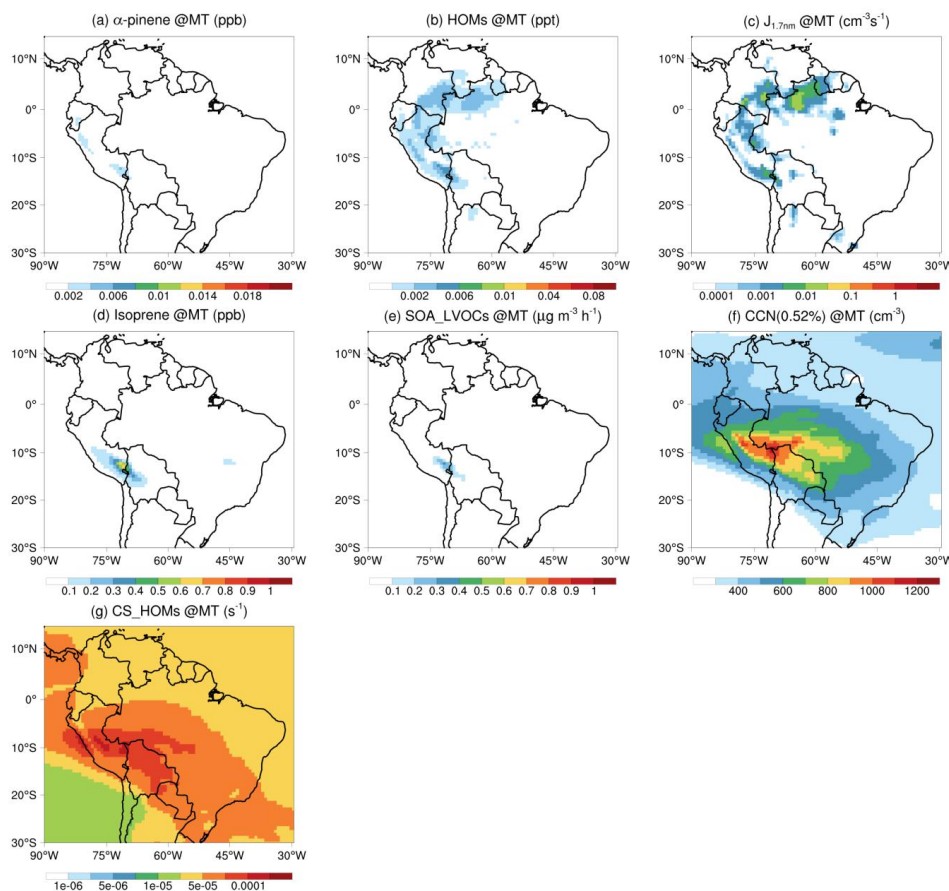


**Figure A10.** Same as Figure 3 (a-g), but for the middle troposphere (MT).





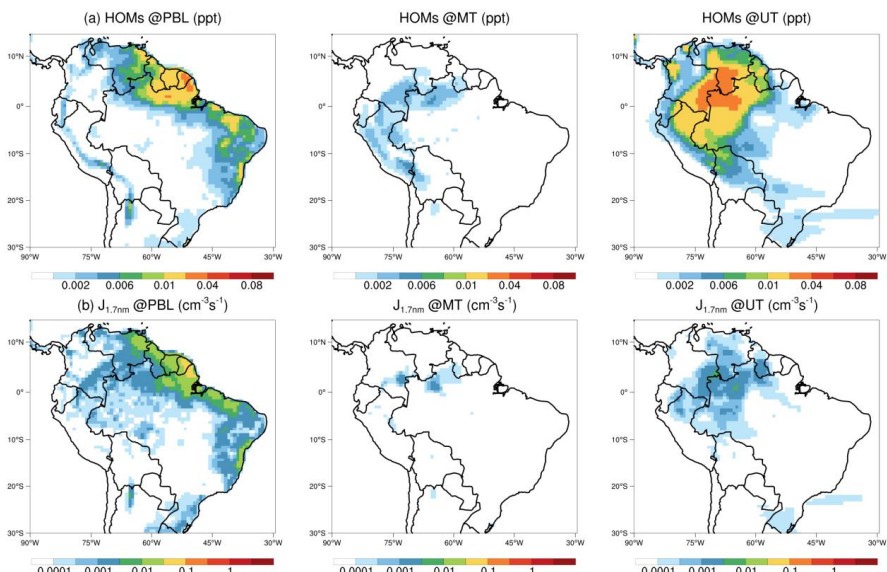

**Figure A11.** Horizontal distribution of (a) HOMs and (b) organic nucleation rate averaged over 1 Sep–1 Oct 2014 from the BNUnoT case.


The simulated $SO_2$ concentration at the location of the ATTO site (Fig. A5) showed a fair agreement with the observations in the free troposphere of the Central Amazon (Andreae & Andreae, 1988; Paralovo et al., 2019; Ramsay et al., 2020), which is an important prerequisite for an accurate simulation of the $H_2SO_4$-$H_2O$ binary nucleation in this model (Wexler et al., 1994). The modeled $SO_2$ concentration in the PBL, especially near the ground surface, was higher than the observations (Ramsay et al., 2020), which could be related to several factors, e.g. overestimated $SO_2$ emission (Andreae, 2019) and/or inadequate scavenging (Hardacre et al., 2021), and requires further investigations. The influence of the $SO_2$ overestimation in the PBL on the simulated aerosol concentration was examined by conducting a sensitivity study, namely, BASE_$SO_2$_constrain, where all the settings were the same as the BASE case except that the $SO_2$ concentration in the PBL was fixed to 30 ppt in accordance with the lower end of the range of published measurements during the dry season. The simulation results show that the difference in the aerosol concentration within the PBL between BASE and BASE_$SO_2$_constrain is minor (Fig. A6), which indicates an insignificant influence of the $SO_2$ overestimation in the PBL on the simulated aerosol concentration.







**Table A4.** Averaged values of modeled and observed aerosol particle number concentration.

|  | CN (cm$^{-3}$) | | | CCN(0.52%) (cm$^{-3}$) | | |
|---|---|---|---|---|---|---|
|  | PBL* | MT* | UT* | PBL* | MT* | UT* |
| Observation (Andreae et al., 2018) | 1650±1030 | 2130±3070 | 7700±7970 | 880±630 | 410±150 | 840±440 |
| BASE | 2230 | 2490 | 6130 | 800 | 490 | 350 |
| BASEnoNUC | 2150 | 1530 | 390 | 800 | 440 | 160 |
| CTRL | 2700 | 2220 | 6010 | 1100 | 580 | 750 |
| OCD | 2390 | 2140 | 3900 | 1090 | 570 | 660 |
| BNUnoT | 3280 | 2180 | 3980 | 1170 | 580 | 660 |
| OCDnoT | 2660 | 2430 | 5950 | 1370 | 540 | 460 |
| NoOH | 2470 | 2170 | 4000 | 1090 | 580 | 670 |

*PBL, MT, and UT are defined as the altitude range of 0–4 km, 5–8 km, 9–15 km, respectively.

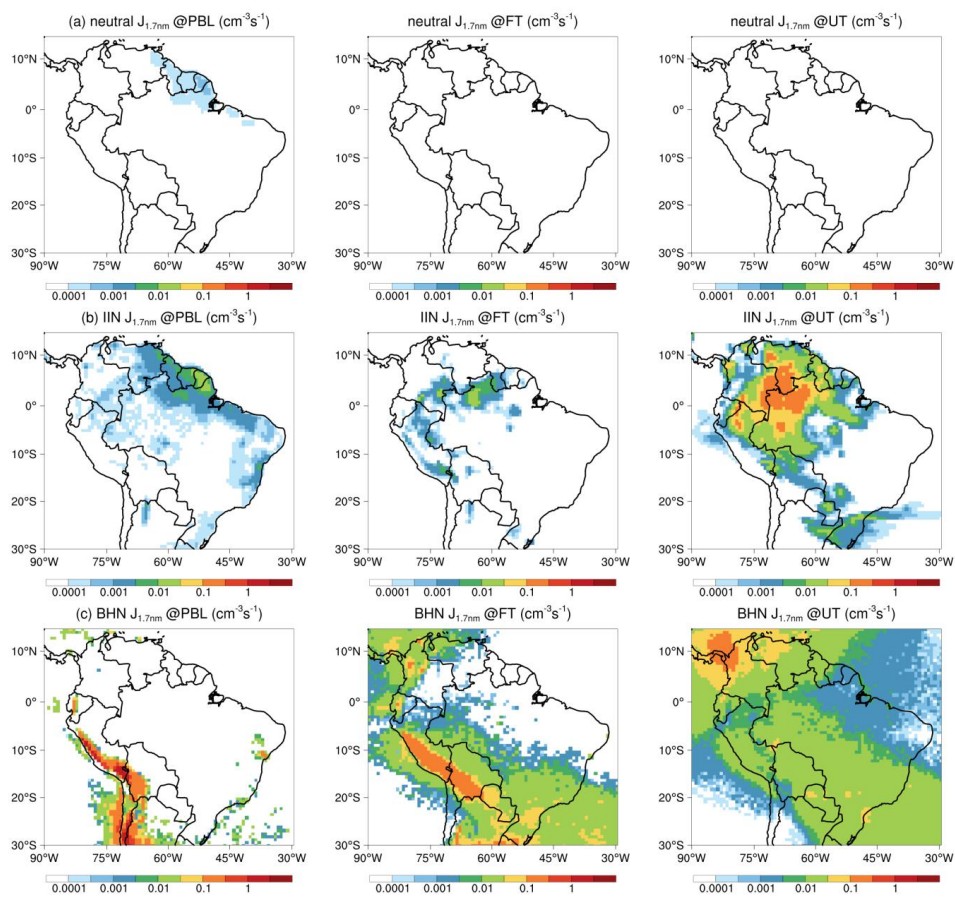

**Figure A12.** Horizontal distribution of (a) neutral organic nucleation rate, (b) ion-induced organic nucleation

rate, and (c) H₂SO₄-H₂O binary homogeneous nucleation rate at the planetary boundary layer (PBL, left panel),

middle troposphere (MT, middle panel), and upper troposphere (UT, right panel) averaged over 1 Sep–1 Oct

2014 from the CTRL case.



In this study, the $H_2SO_4$-$H_2O$ binary nucleation mechanism was simulated as it has been widely used for describing inorganic nucleation in the free troposphere (Cui et al., 2014; Gordon et al., 2016; Zhu & Penner, 2019). As shown from the nucleation rate in Fig. A12, $H_2SO_4$-$H_2O$ binary nucleation mainly occurs in the free troposphere, which is consistent with the vertical distribution of binary nucleation simulated for the Amazon region in Zhao et al. (2020). The $H_2SO_4$-$H_2O$ binary nucleation causes a CN increase of over 3000 $cm^{-3}$ in the UT under sufficient particle condensational growth as approximately estimated from the difference between OCD and BASEnoNUC (Table A4). It is of a comparable magnitude to the CN increase of 2100 $cm^{-3}$ by organic nucleation. A higher rate of the $H_2SO_4$-$H_2O$ binary nucleation over the organic nucleation was also found by Zhao et al. (2020) in the Amazon from 9 to 13 km altitude but the overall $H_2SO_4$-$H_2O$ binary nucleation in the UT was insignificant in Zhao et al. (2020) which is different from the result in this study. This is expected as the result in Zhao et al. (2020) was for a low-$SO_2$ area and there was competition for $H_2SO_4$ by other $H_2SO_4$-involving inorganic nucleation processes in Zhao et al. (2020). In a global simulation where the inorganic nucleation was represented only by the $H_2SO_4$-$H_2O$ binary nucleation, the column-integrated $H_2SO_4$-$H_2O$ binary nucleation in the Amazon is of the same magnitude as but somewhat lower than the organic nucleation (Zhu & Penner, 2019). Considering the $H_2SO_4$-$H_2O$ binary nucleation occurs mainly in the upper troposphere and the organic nucleation in Zhu & Penner (2019) includes the hetero-molecular organic nucleation, the relative importance of $H_2SO_4$-$H_2O$ binary nucleation to pure organic nucleation in the UT should be greater than shown in the column-integrated results. Therefore the simulated $H_2SO_4$-$H_2O$ binary nucleation in this study should be generally reasonable.

The model can generally reproduce the vertical distribution of CN (Fig. 1) as described in Section 2.1, yet a systematic overestimation of CN exists in model simulations below 5 km. This could be associated with uncertainties in the fire emission inventories (Andreae, 2019), as aerosols near the ground surface during the Amazon dry season are overwhelmingly influenced by persistent biomass burning (Andreae et al., 2015). In addition, the comparison of a grid-average value in the model with an observation on a spot may also contribute to the discrepancy. To make the CCN comparison, the modeled aerosols of a size consistent with that of the observed CCN(0.52%) were used. The cut-off size of the CCN(0.52%) was calculated to be approximately 90 nm in diameter (Su et al., 2010), based on an observed average hygroscopicity value of 0.12 derived from the aerosol component observations. This hygroscopicity level is close to those of organic aerosols (Petters and Kreidenweis, 2007), which is expected since organic aerosols dominate the aerosol components in this area (Andreae et al., 2018). As the main aerosol component (i.e., organic aerosols) can be well reproduced by the developed model version (Fig. 2b), it is justifiable to use this observed hygroscopicity to calculate the CCN size in models. The comparison shows that the CCN number was underestimated by the BASE case. The model underestimation of CCN number in the UT reaches -58% as shown in Section 2.1. Compared with the UT, the biases in CCN below 4 km are much smaller and lie around -16%. The negative biases in CCN number in both the UT and PBL are however corrected in the CTRL case, suggesting the lack of SOA production and inadequate particle growth in the BASE case as the reason for the CCN underestimation.


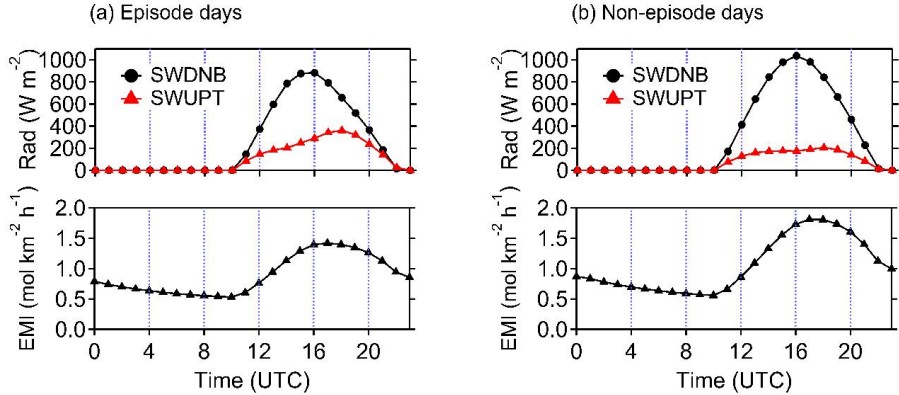

**Figure A13.** The diurnal variation of radiation flux (upward shorwave radiation at the top of the atmosphere (SWUPT) and downward shortwave radiation at the surface (SWDNB)) and α-pinene emission rate for the average of (a) upper troposphere (UT) biogenic SOA episode days and (b) non-episode days.

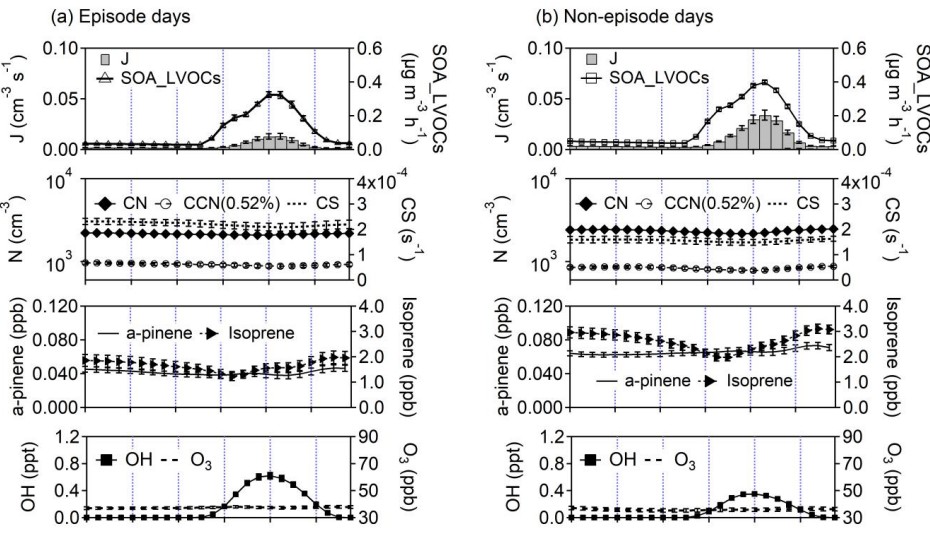


**Figure A14.** Same as Fig. 4b and Fig. 4c, but for the planetary boundary layer (PBL).



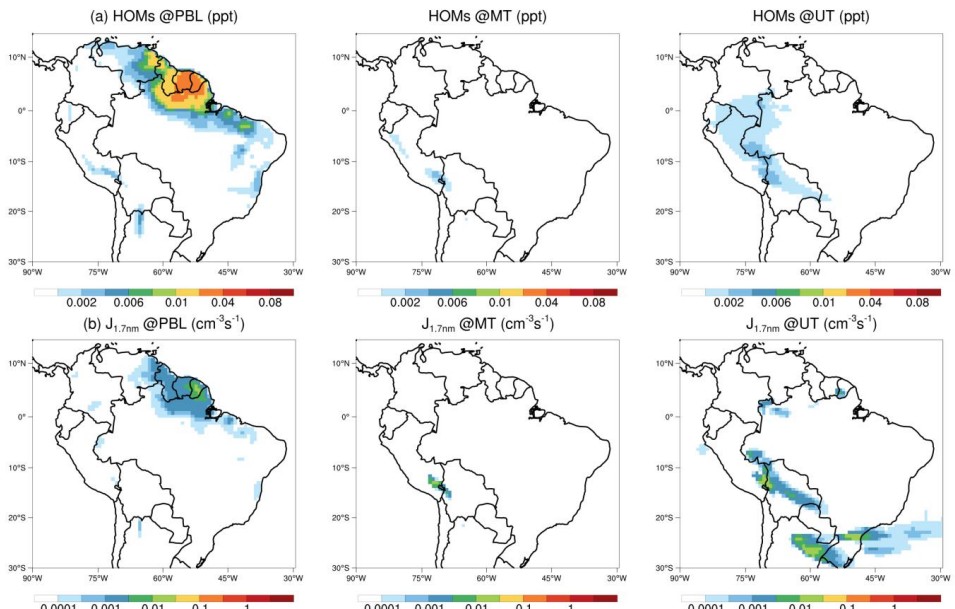

**Figure A15.** Horizontal distribution of (a) HOMs and (b) organic nucleation rate averaged over 1 Sep–1 Oct 2014 from the NoOH case.


### A4. Sensitivity simulations of LVOCs condensation

Figure 2b compares the simulated OA mass from CTRL and BASE with the observed data. The BASE case using the MOSAIC aerosol module and FINN biomass burning emission inventory shows a reasonable performance of OA representation in the PBL, which was also confirmed by previous evaluations for this region

(Archer-Nicholls et al., 2015; Wang et al., 2016b); however, the OA in the UT is significantly underestimated. This negative bias of OA mass in the UT in the BASE case is greatly improved in the CTRL case by considering the organic aerosol processes driven by the biogenic precursors, among which the LVOCs condensation plays a dominant role (Fig. 2a).

To further examine the uncertainty of the LVOCs condensation in terms of the LVOC yields, sensitivity

simulations regarding the temperature dependence of the LVOC yields were performed. The OCDnoT case adopted a bulk assumption of a yield of 13% from monoterpene oxidation and 3% from isoprene oxidation (OCDnoT), as suggested by Scott et al. (2014); while in the CTRL case, the temperature dependence of LVOC yields based on an α-pinene oxidation experiment (Saathoff et al., 2009) was applied to the LVOC yields. The OCDnoT case produces a larger amount of boundary layer OA than the CTRL case, causing a higher bias in the

model compared with the observations and suggesting an excessive SOA production. A significant difference between the environment where the LVOC yields were originally based (Kroll et al., 2005) and the region investigated here may be the reason for the poor performance of the bulk yields in the studied area, as the temperature in previous applications is much lower than the tropical forest boundary layer conditions. On the



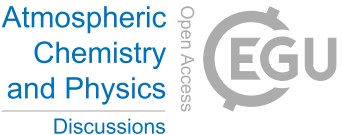

other end, the temperature difference could also explain the underestimation of OA mass with bulk yields in the

OCDnoT for the UT (Fig. 2b), where the temperature is far below the freezing point (Fig. A2). With the temperature dependence correction, i.e., LVOC yields increasing with colder temperature, the OA underestimation in the UT in the OCDnoT case can be effectively corrected in the CTRL case.