# Peer review of "Strong particle production and condensational growth in the upper troposphere sustained by biogenic VOCs from the canopy of the Amazon Basin"

_Atmospheric Chemistry and Physics, 2022_

## Author Comment (AC1)

Dear Referee #1,

Thank you for your comments and suggestions. Below I provide a point-by-point response to individual comments (referee comments and suggestions are in italics, responses and revisions are in plain font; revised sections in the manuscript text in response to the comments are marked with red color).

**Comments and suggestions from Referee #1**

**Comment 1:**

*Why did this manuscript need to be as short as it is? I would appreciate at least the key methods being moved to the main text. It was strange to need to go to the SI to find out what the "binary nucleation" scheme was given that this is a nucleation & growth manuscript (and the findings greatly depend on the initial nucleation scheme).*

**Responses and revisions 1:**

We thank the referee for the kind suggestion. As already mentioned by Dr. Eimear Dunne (referee #2), this manuscript has been for 'ACP Letters', which requires the main text to be short and the applied methods to be in the form of an appendix after the main text. To address the problem the referee put forward, we have added the literature citation to the main text when mentioning "binary nucleation" so that readers can get a basic idea of what binary nucleation scheme being used when reading the main text, and the detailed description of the binary nucleation scheme is provided in the Appendix. Line 83, '- BASE, the default WRF-Chem simulation with $H_2SO_4$-$H_2O$ binary nucleation (Wexler et al., 1994) and without biogenic nucleation or condensation;'

***Comment 2:***

*I believe that organics being the key missing ingredient in growing particles to CCN sizes in the UT above the Amazon is likely a robust finding (achieved through closure for both OA mass and CCN number). However, I do not believe that the findings about the role of organics-only nucleation are robust, and I believe these findings are overstated. The base WRF-Chem simulations had only a single, very old binary nucleation "scheme" (Wexler 1994 only gives the critical H2SO4 concentration required for nucleation to initiate, one still needs to assume a nucleation rate!). No recent binary or ternary schemes (e.g., Dunne) were investigated, nor organic-sulfuric nucleation (e.g., Riccobono with the Yu temperature correction). How can we say with confidence that organics-only nucleation dominates in the UT above the Amazon? We can't. The fractional increases in the CN and CCN concentrations due to the organics-only scheme is entirely dependent on Wexler 1994 being the starting point. Please soften these findings to have the effect of "we find the organics-only nucleation can reproduce CN and CCN concentrations, but in the absence of testing other schemes, we cannot say definitively if organics-only nucleation dominates in the UT above the Amazon."*

**Responses and revisions 2:**

We thank the referee for the insightful comment and suggestion. We definitely agree with the referee that conclusions can not be drawn in this study as to whether or not pure organic nucleation is the dominant nucleation mechanism in the UT. Actually, based on the simulation results, the increase in CN concentration due to organic nucleation is 2100 $cm^{-3}$ which is lower than the CN increase of over 3000 $cm^{-3}$ due to $H_2SO_4$-$H_2O$ binary nucleation in the UT as approximately estimated from the difference between OCD and BASEnoNUC (Table A4; Line 712). Thus, we by no means intended to convey the idea that pure organic nucleation dominates in the UT. The effect of organic nucleation on CN in the UT, i.e. an increase of 2100 $cm^{-3}$, is over one quarter of the observed total CN concentration (7700 $cm^{-3}$, Table A4), and we described this effect as "strong particle production (Line 1)", "play important roles in maintaining the particle population and size distribution in the UT (Line 128) ", and "effectively increases the CN number by replenishing new nano-sized particles (Line 129)".

The sentence in Line 69 that "The organic nucleation mechanism in this study focuses on pure organic nucleation, ...., as it was found dominant among organic nucleation pathways in the Amazon (Zhu & Penner, 2019)" could be misleading. This is a result of Zhu & Penner (2019, Figure 3) for comparison among organics-involved nucleation pathways instead of a result of this study. To avoid misleading the readers, we accept the referee's suggestion and have added a clarification about the role of pure organic nucleation:

Line 220, "Note that although pure organic nucleation contributes importantly to the aerosol population in the UT, the relative roles of pure organic nucleation and other nucleation mechanisms, such as ternary and ion-induced inorganic nucleation (Napari et al., 2002; Yu et al., 2008), in the UT aerosol production remain to be investigated with a comprehensive consideration of nucleation parametrizations, e.g. those in Dunne et al. (2016) and Riccobono et al. (2014)."

And the short summary of the manuscript has been changed from

"we show that the UT aerosol formation triggered by biogenic organics shapes the UT aerosols, and organic condensation is key for UT CCN production." to

"we show strong aerosol nucleation and condensation in the UT triggered by biogenic organics, and organic condensation is key for UT CCN production."

*Comment 3:*

*Throughout the manuscript, values of concentrations, rates-per-volume, and the condensation sink are given without stating if the values are for local temperature and pressure or at STP. This information is critical since a lot of the values are for 8 km, far from STP (and sometimes are put next to mixing ratios that do not depend on T&P).*

**Responses and revisions 3:**

Thanks for pointing out this problem. The values of the concentrations, rates-per-volume, and condensation sink in this manuscript are values at STP. This information has been added now.

Figure 1, line 102, "The aerosol concentrations are at standard temperature and pressure (STP; 273.15 K and 1000 hPa)."

Figure 2, line 136, "The OA production rate and aerosol mass concentrations are at STP."

Figure 3, line 152, "Spatial distribution of (a) α-pinene, (b) HOMs, (c) organic nucleation rate, (d) isoprene, (e) SOA production rate by LVOCs, (f) CCN at 0.52% supersaturation, and (g) condensation sink of HOMs ..., all at STP."

Figure 4, line 187, "The concentrations of gases and aerosols, the production rates, and the condensation sink are normalized to STP."

Figure A3, line 652, "Comparison of (a) $O_3$ mixing ratio, and (b) black carbon (BC) mass concentration ..., all at STP."

Figure A4, line 656, "Simulated (a) vertical profiles and (b) time series of α-pinene (API), β-pinene (BPI), and isoprene (ISO) mixing ratios (STP)"

Figure A5, line 667, "Simulated vertical profile of the $SO_2$ mixing ratio (STP) at the location of ATTO."

Figure A6, line 671, "Vertical profiles of the simulated number concentrations (STP) of CN and CCN..."

Figure A7, line 677, "The particle size distributions are normalized to STP."

Figure A8, line 681, "The concentrations are at STP."

Figure A11, line 689, "Horizontal distribution of (a) HOMs and (b) organic nucleation rate..., all at STP."

Figure A12, line 713, "The nucleation rates are for STP."

Figure A15, line 760, "Horizontal distribution of (a) HOMs and (b) organic nucleation rate..., all at STP."

Table A4, "CN ($cm^{-3}$, STP) |      CCN(0.52%) ($cm^{-3}$, STP)"

Line 640, "The gas and aerosol concentrations have been normalized to standard temperature and pressure (STP)."

Line 660, "The modeled gas and aerosol concentrations are values at STP, consistent with the observed data."

*Comment 4:*

*L190: 1e^-3 cm^-3 s^-1. Is this supposed to be 1x10^-3 cm^-3 s^-1 (or equivalently 1E-3 cm^-3 s^-1)? Very weird to use base e for scientific notation.*

**Responses and revisions 4:**

Thanks for pointing out this error. Now we have changed 1 e^-3 to "$1\times10^{-3}$" (line 191).

---

## Author Comment (AC2)

Dear Dr. Eimear Dunne,

Thank you for your comments and suggestions. Below I provide a point-by-point response to individual comments (referee comments and suggestions are in italics, responses and revisions are in plain font; revised sections in the manuscript text in response to the comments are marked with red color).

**Comments and suggestions from Dr. Eimear Dunne**

**Comment 1:**

*The value 1 e^{-3} on line 190 is definitely wrong somehow, and probably meant to read 1E-3 or equivalent*

**Responses and revisions 1:**

Thanks for pointing out this error. Now we have changed 1 e^{-3} to "$1 \times 10^{-3}$" (line 191).

*Comment 2:*

*Figure A4 (a) would benefit from having a log scale on the x-axis, maybe as an extra panel*

**Responses and revisions 2:**

Thanks for the suggestion. We have added an extra panel using a log scale to Figure A4 (a), so that the vertical patterns in the middle and upper troposphere are clearer now:

[Figure]

**Figure A4.** Simulated (a) vertical profiles and (b) time series of α-pinene (API), β-pinene (BPI), and isoprene (ISO) mixing ratios (STP) at the location of ATTO. The embedded figure in (a) is the same as the outer figure but on a log scale.

***Comment 3:***

*It is claimed that Figure A5 shows a fair agreement with observations, but A5 only shows simulated values - please support your claim by also plotting the relevant observations*

**Responses and revisions 3:**

Thanks for the comment. Here we did not make a point-to-point comparison between simulated and observed $SO_2$ due to the unavailability of the aircraft observation of $SO_2$ during the ACRIDICON-CHUVA campaign. The observed $SO_2$ concentrations used for the comparison are from published papers. Andreae & Andreae (1988) observed an $SO_2$ concentration of 18 ppt in the free troposphere (FT) based on aircraft measurements over the Amazon Basin during July and August, which represents a background condition with little impact from anthropogenic plumes. The simulated $SO_2$ concentration in this study is generally around 21 ppt in the free troposphere (except the high value at ~6 km) and decreases near the tropopause. The simulated result is 3 ppt higher than the observation but basically captures the magnitude of the $SO_2$ in the free troposphere. Therefore we consider this simulation result is in fair agreement with observations. We accept the referee's suggestion and have now explicitly added the observed results from Andreae & Andreae (1988) in the manuscript so that the comparison is clearer:

Line 691:

"The simulated $SO_2$ concentration of around 21 ppt throughout most of the FT at the location of the ATTO site (Fig. A5) is in fair agreement with an observed background $SO_2$ concentration of 18 ppt in the FT over the Central Amazon (Andreae & Andreae, 1988)."

As for the $SO_2$ in the planetary boundary layer (PBL), Andreae & Andreae (1988) observed an $SO_2$ concentration of 27 ppt in the PBL from aircraft measurements; Ramsay et al. (2020) observed an $SO_2$ concentration of 80 ppt at ATTO at 60 m. The simulated $SO_2$ concentration is around 68 ppt at the higher part of the PBL (above 300 m) but shows a high value of 130 ppt near the ground surface. Therefore the simulated $SO_2$ in the PBL is relatively high compared to observations, especially for near surface. Then we conducted a sensitivity study to constrain the whole PBL $SO_2$ concentration to 30 ppt to test the influence of the $SO_2$ overestimation in PBL on the aerosol simulation. The sensitivity study (Fig. A6) shows that the $SO_2$ overestimation in PBL only causes a minor difference in the aerosol concentration and does not substantially affect the aerosol simulation results in this study.

For the $SO_2$ comparison in the PBL, we have also added the observation results to make the comparison clearer:

Line 694:

"Compared to the observed $SO_2$ concentration of 27 ppt in the PBL (Andreae & Andreae, 1988) and 80 ppt near the ground surface (Ramsay et al., 2020), the modeled $SO_2$ concentration in the PBL, especially near the ground surface, was relatively higher (Fig. A5)."

***Comment 4:***

*With those issues out of the way, it's time to address my main concern: the Wexler et al. (1994) nucleation parameterisation. I understand why the paper did not update the default nucleation scheme in WRF-Chem; after all, the authors were already implementing a new nucleation scheme, and it makes sense to compare it to the existing set-up. This is especially true for a model like WRF, where there are so many different configurations available. So I don't think it would be even slightly reasonable to suggest rejecting the paper on these grounds, but I do think that there needs to be more acknowledgement of how a nucleation parameterisation affects CCN in the simulated upper troposphere. The Wexler et al. (1994) publication explicitly states:*

*"The number of particles produced by this nucleation operator is somewhat arbitrary [...] Any error produced by this treatment is mitigated in the SoCAB because the vast majority of the aerosol loading is due to primary emission and condensation of secondary organic compounds. In locations where nucleation is more significant, this treatment may not be sufficiently accurate."*

*In the upper troposphere, nucleation is the only local source of aerosols. If the real conditions being simulated are actually saturated with respect to freshly nucleated particles, and the default nucleation parameterisation under-predicts the true nucleation rate significantly, then changing to any parameterisation which predicts a value close to the saturation limit will improve predictions; but it cannot then be concluded that the nucleation pathway is the dominant one in that region, even if that is the case in reality.*

*I would agree with Referee #1 that the conclusions ought to be softened. If any nucleation scheme that was known to be more robust in the UT had been used, I would have been happy to accept the conclusions as they stand. However, I would also be happy to discuss implementing the Dunne et al. (2016) scheme in WRF-Chem with the authors, if they would be interested in a future collaboration where their stronger conclusions might yet be validated!*

**Responses and revisions 4:**

We thank the referee for the insightful comment and patient explanation. We agree that in this study we can not conclude a dominant role of pure organic nucleation without testing other nucleation mechanisms. In this manuscript, we describe the effect of organic nucleation on CN in the UT (i.e. an increase of 2100 cm$^{-3}$, over one quarter of the observed total CN concentration of 7700 cm$^{-3}$, Table A4) as "strong particle production (Line 1)", "play important roles in maintaining the particle population and size distribution in the UT (Line 128) ", and "effectively increases the CN number by replenishing new nano-sized particles (Line 129)" instead of concluding it as a dominant mechanism. We realize that the sentence on Line 69 that "The organic nucleation mechanism in this study focuses on pure organic nucleation, ...., as it was found dominant among organic nucleation pathways in the Amazon (Zhu & Penner, 2019)" could be misleading. This is not the conclusion of this study but a result of Zhu & Penner (2019, Figure 3) who made comparisons among organics-involved nucleation pathways. Also, the short summary of the manuscript was in a relatively strong tone and has been softened from

"we show that the UT aerosol formation triggered by biogenic organics shapes the UT aerosols, and organic condensation is key for UT CCN production." to

"we show strong aerosol nucleation and condensation in the UT triggered by biogenic organics, and organic condensation is key for UT CCN production."

To address the referee's concern about the simulation performance of the 'Wexler 1994' scheme, we compared the binary nucleation simulation using the Wexler 1994 scheme in this study to the simulations in other studies. The comparison shows that the aerosol production by Wexler 1994 scheme generally agrees with the simulation by other schemes both in vertical distribution and in magnitude:

Line 716, "As shown from the nucleation rate in Fig. A12, H$_2$SO$_4$-H$_2$O binary nucleation mainly occurs in the free troposphere, which is consistent with the vertical distribution of binary nucleation

simulated for the Amazon region in Zhao et al. (2020). The $H_2SO_4$-$H_2O$ binary nucleation causes a CN increase of over 3000 cm$^{-3}$ in the UT under sufficient particle condensational growth as approximately estimated from the difference between OCD and BASEnoNUC (Table A4). It is of a comparable magnitude to the CN increase of 2100 cm$^{-3}$ by organic nucleation. A higher rate of the $H_2SO_4$-$H_2O$ binary nucleation over the organic nucleation was also found by Zhao et al. (2020) in the Amazon from 9 to 13 km altitude but the overall $H_2SO_4$-$H_2O$ binary nucleation in the UT was insignificant in Zhao et al. (2020), which is different from the result in this study. This is expected as the result in Zhao et al. (2020) was for a low-$SO_2$ area and there was competition for $H_2SO_4$ by other $H_2SO_4$-involving nucleation processes in Zhao et al. (2020). In a global simulation where the inorganic nucleation was represented only by the $H_2SO_4$-$H_2O$ binary nucleation, the column-integrated $H_2SO_4$-$H_2O$ binary nucleation in the Amazon is of the same magnitude as but somewhat lower than the organic nucleation (Zhu & Penner, 2019). Considering the $H_2SO_4$-$H_2O$ binary nucleation occurs mainly in the upper troposphere and the organic nucleation in Zhu & Penner (2019) includes the hetero-molecular organic nucleation, the relative importance of $H_2SO_4$-$H_2O$ binary nucleation to pure organic nucleation in the UT should be greater than shown in the column-integrated results. Therefore the simulated $H_2SO_4$-$H_2O$ binary nucleation in this study should be generally reasonable.

For the question of how nucleation parameterisation affects CCN in the upper troposphere, we found the simulations in Westervelt et al. (2014) based on a global model GEOS-Chem-TOMAS can provide some hints. The sensitivity simulations in Westervelt et al. (2014; Figure 3) show that the CCN in the upper troposphere did not vary substantially when the nucleation scheme changed from binary nucleation to ternary nucleation or even when a tuning factor of 10$^{-5}$ was applied to the nucleation rate. Based on the simulation results, the nucleation parameterisation did not seem to affect the upper tropospheric CCN in a significant way. However, these simulations were conducted by global models while sensitivity simulations by regional models where convective transport can be better resolved are still lacking. It's worthwhile to conduct such a sensitivity study, and we highly look forward to a corporation with the referee in the future. Here we accept the referee's suggestion and have added acknowledgment of the need to further investigate other nucleation schemes in order to understand their relative roles.

Line 220, "Note that although pure organic nucleation contributes importantly to the aerosol population in the UT, the relative roles of pure organic nucleation and other nucleation mechanisms, such as ternary and ion-induced inorganic nucleation (Napari et al., 2002; Yu et al., 2008), in the UT aerosol production remain to be investigated with a comprehensive consideration of nucleation parametrizations, e.g. those in Dunne et al. (2016) and Riccobono et al. (2014)."

Westervelt, D. M., Pierce, J. R., and Adams, P. J.: Analysis of feedbacks between nucleation rate, survival probability and cloud condensation nuclei formation, Atmos. Chem. Phys., 14, 5577–5597, https://doi.org/10.5194/acp-14-5577-2014, 2014.